# Effect of Protein-Fortified Diet on Nitrogen Balance in Critically Ill Patients: Results from the OPINiB Trial

**DOI:** 10.3390/nu11050972

**Published:** 2019-04-28

**Authors:** Matteo Danielis, Giulia Lorenzoni, Danila Azzolina, Anna Iacobucci, Omar Trombini, Amato De Monte, Dario Gregori, Fabio Beltrame

**Affiliations:** 1Department of Anaesthesia and Intensive Care, Azienda Sanitaria Universitaria Integrata di Udine, Udine 33100, Italy; matteo.danielis@asuiud.sanita.fvg.it (M.D.); anna.iacobucci@asuiud.sanita.fvg.it (A.I.); omar.trombini@asuiud.sanita.fvg.it (O.T.); amato.demonte@asuiud.sanita.fvg.it (A.D.M.); fabio.beltrame@asuiud.sanita.fvg.it (F.B.); 2Unit of Biostatistics, Epidemiology and Public Health, Department of Cardiac, Thoracic, Vascular Sciences and Public Health, University of Padova, Padova 35131, Italy; giulia.lorenzoni@unipd.it (G.L.); danila.azzolina@unipd.it (D.A.)

**Keywords:** intensive care unit, nutritional therapy, protein intake, nitrogen balance, randomized controlled trial, energy intake

## Abstract

Nitrogen balance (NB) is considered a good marker of adequate protein intake and it has been suggested to be a good predictor of patients’ health outcomes. However, in literature, there is a lack of large randomized trials examining NB-guided protein intake in patients in intensive care units (ICUs). A randomized controlled trial enrolling patients admitted to ICU was done to compare changes in NB. Participants were randomized to a standard or protein-fortified diet (protein intake of 1.8 g/kg/day according to the guidelines of the Society of Critical Care Medicine and the American Society for Parenteral and Enteral Nutrition). The primary endpoint was represented by the NB on Day 1, 3, and study exit. Forty patients were enrolled in the study (19 in the protein-fortified group). The longitudinal analysis showed that, on Day 3, patients randomized to the protein-fortified diet were more likely (*p* < 0.001) to present better NB (at 3 days, patients in the protein-fortified diet were estimated to have a nitrate value of 5.22 g more than patients in the standard diet, 95% CI 3.86–6.58). The protein-fortified diet was found to be significantly and directly associated with changes in NB in critically ill patients admitted to ICU.

## 1. Introduction

The nitrogen balance (NB) reflects the gain or loss of total body protein and is represented by the difference between dietary nitrogen intake and nitrogen losses (represented by nitrogen excreted in the urine and by insensible losses via the skin and gastrointestinal tract). It is considered an easy and inexpensive method that is useful for the assessment of protein turnover and the effectiveness of nutritional therapy [1].

The occurrence of stressful conditions (i.e., organ failure, trauma, or sepsis) might result in an impairment of NB. The hypermetabolism and whole-body protein catabolism resulting from a stressful event in critically ill patients might lead to a negative NB [1]. Critically ill patients admitted to intensive care units (ICUs) usually exhibit intense protein catabolism, and this effect is undesired since protein losses are associated with increased morbidity and mortality [2].

The NB is considered a good marker of adequate protein intake and it has been suggested to be a good predictor of patients’ health outcomes [3]. However, in literature, there is a lack of large randomized trials examining NB-guided protein intake upon clinical outcomes for patients admitted to ICU. Guidelines on nutritional therapy of critically ill patients [4] recommend a high-protein diet in ICU, and protein recommendations should be adjusted using NB for each patient as a predictor of protein intake and nitrogen losses to achieve nitrogen equilibrium [3]. In addition to that, it is still unclear whether the recommended intake of proteins is adequate to improve nitrogen losses of critically ill patients. Only a few studies have been conducted in this field [5,6,7]. Such investigations seem to suggest that an increased protein intake might be associated with improvement in NB and might be beneficial for critically ill patients, but results are inconclusive [3]. A prospective observational cohort study on 113 ICU patients has suggested that a protein intake of 1.5 g/kg/day vs. a protein intake of 1.1 g/kg/day or 0.8 g/kg/day led to a significant improvement of the NB and ICU mortality [8]. Conversely, a randomized controlled trial on 119 patients in ICU reported that NB was significantly better in the intervention group (administered with 1.2 g/kg of amino acids) compared to the control group on day 3 but not on day 7 from admission and no differences were detected in the mortality rate [5].

Since NB (determinants, consequences, optimal intake) in ICU patients has not yet been investigated, despite its relevance for critically ill patients’ health outcomes, it is crucial to conduct further research in this field.

The present study aimed to compare changes in NB (considered as a marker of nitrogen loss) in critically ill patients admitted to ICU and randomized to a standard and a protein-fortified diet.

## 2. Materials and Methods

### 2.1. Study Design

Randomized controlled trial (Optimizing Protein Intake and Nitrogen Balance -OPINiB- study, Trial Registration: ClinicalTrials.gov NCT02990065) that enrolled adult patients (ages >18 years) admitted to the Department of Anaesthesia and Intensive Care, Azienda Universitaria Integrata di Udine (Italy), who were undergoing mechanical ventilation within 12 h from admission, and receiving parenteral or enteral nutrition. Subjects were not eligible to the study if they were malnourished (body mass index (BMI) <18.5 or ≥30 kg/m^2^), pregnant, terminally ill, and affected by acute/chronic renal or hepatic failure.

### 2.2. Experimental Procedure

Once admitted to the ICU and assessed for eligibility by a physician, patients were randomized (using a computer-generated algorithm with an allocation rate of 1:1) to a standard or protein-fortified diet. The protein-fortified diet consisted of protein intake of 1.8 g/kg/day according to the most recent guidelines in the field [4] recommending a protein intake between 1.2 and 2 g/kg/day. The energy requirements (other than proteins) were calculated according to the resting energy expenditure (REE) formula [9], accounting for patients’ age (below or above 60 years). The total energy counting for patients in the protein-fortified group was then calculated as follows: REE × 0.5 + Protein Intake. The standard diet consisted of an energy goal set at 20–25 Kcal/kg/day, according to international guidelines [4]. Details of the study protocol are given elsewhere [10].

Each patient enrolled in the study could undergo enteral and/or parenteral nutrition according to the clinical judgement and guidelines in the field. The nutritional support (enteral and/or parenteral) was set up in order to comply with the energy requirements of each subject. The nutritional composition of enteral nutrition and parenteral nutrition for the standard diet was as follows: for enteral nutrition, protein (16%), carbs/dextrose (35%), fats (49%); for parenteral nutrition, protein (16%), carbs/dextrose (49%), fats (35%). The nutritional composition of enteral nutrition and parenteral nutrition for the protein-fortified diet was as follows: for enteral nutrition, protein (21%), carbs/dextrose (31%), fats (46%), fiber (2%); for parenteral nutrition, protein (22%), carbs/dextrose (40%), fats (38%). The proportion of protein administered to the subjects allocated to the protein-fortified diet was specific for each patient according to her/his energy requirements, the percentages reported are average values.

### 2.3. Data Collection

At time of hospital admission (Day 0), subjects’ demographic (age, gender), clinical (diagnosis, clinical history, illness severity using the Acute Physiology and Chronic Health Evaluation (APACHE) II score, risk of death within the ICU using the Simplified Acute Physiology Score (SAPS)), habits (cigarette smoking), and anthropometric characteristics (weight and height, which were used to calculate the body mass index (BMI) [11]) were assessed. The weight was assessed using the ICU bed weighing system; as per patient height, a trained nurse measured the knee height of each patient using a measurement tape. The knee height was used to estimate the height using the Chumlea equation, which was validated also in the ICU setting [12]. The smoking habit was assessed within the routinely nursing admission assessment, involving patients’ relatives/caregivers.

At midnight of each day, a standard set of biochemical parameters (blood creatinine, 24-h urine urea nitrogen (UUN) excretion, blood urea nitrogen (BUN), glycemia), together with protein and caloric intake, were collected. Blood analyses were carried out using spectrophotometry. NB was calculated daily using the standard formula described by Jivnani et al. [13]. Subjects remained in the study until mechanically ventilated. Withdrawal criteria were as follows: term of mechanical ventilation, onset of acute renal or hepatic failure, transfer to another hospital, death.

REDCap [14], a web-based application for data management, was employed for data collection.

### 2.4. Outcomes

The NB represented the primary endpoint. The occurrence of ICU-acquired skin alterations and creatinine levels represented the secondary endpoints. The endpoints were assessed on Day 1, 3, and study exit [8].

### 2.5. Ethics

Appropriate permissions were obtained from the regional ethics committee of the Friuli Venezia Giulia (CEUR-2016-Sper-066-ASUIUD). Patients (or legally authorized representatives) were required to provide written informed consent before study participation. 

### 2.6. Sample Size

NB was considered as the primary outcome. Thirty-eight patients (19 per arm) were estimated to be necessary to detect a difference of at least −6 g of NB between the two groups for a specified alpha of 0.025 and a power of 0.90 [10].

### 2.7. Statistical Analysis

Descriptive statistics of data according to standard and protein-fortified groups were performed. Data were reported as median (first and third quartiles) for continuous variables, and percentages (absolute numbers) for qualitative variables. Wilcoxon–Kruskal–Wallis test was performed for continuous variables and the Pearson chi-square test for categorical ones.

The same descriptive statistics for the distribution of the endpoints (on Day 1, 3, and study exit) were reported. In this study, *p*-values for differences between groups have been adjusted within days by multiplicity of testing using the Benjamini and Hochberg method [15].

An ordinary least square method (OLS) for continuous outcomes and a logistic regression model for binary outcomes were performed including restricted cubic splines (3 knots) on days in ICU interacting with different diet groups. Both interaction and marginal effects were included in the models. The interaction term between the treatment (standard vs. protein-fortified diet) and time was considered to account for potential different time-dependent patterns of the endpoint of interest in the control and in the experimental group.

The *p*-values related to the effects of secondary outcomes were adjusted considering a multiple testing correction [15].

Computations were performed using R 3.3.3 [16] with rms [17] and ggplot2 [18] packages.

## 3. Results

Forty patients were enrolled in the study (19 allocated in the protein-fortified diet group, Figure 1) between January and May 2017. They were followed up for a median of 7 days (5–13.25, I–III quartile).

The patients’ characteristics at the time of ICU admission (Day 0), according to allocation to the protein-fortified and standard diet, are presented in Table 1. No significant differences were found in any one of the characteristics of the study subjects randomized to the standard and protein-fortified diet (Table 1).

Protein and energy intakes on Day 1, Day 3, and study exit in the control and intervention groups are shown in Figure 2 and Figure 3, respectively. 

Protein intake was significantly higher (*p* < 0.001) in the intervention compared to the control group at each time point considered. Conversely, there was no significant difference between the intervention and control groups in the energy intake, except for Day 1 (*p*-value 0.023, a median of 920 kcal in the intervention group and 650 kcal in the control group).

Table 2 presents the distribution of primary and secondary endpoints on Day 1, Day 3, and study exit according to patients’ allocation. 

NB was found to be significantly better in the protein-fortified diet group compared to the standard diet group at each time point analyzed (*p* < 0.001 on Day 1, *p* = 0.003 on Day 3, *p* < 0.001 on study exit). No significant differences were found in the distribution of the secondary endpoints, except for the rate of ICU-acquired skin alterations, which was found to be higher among the standard diet group compared to the protein-fortified ones at study exit (38% vs. 5%, respectively, *p*-value 0.013).

The longitudinal analysis of primary endpoint (Figure 4), on Day 3, showed that patients randomized to the protein-fortified diet were significantly (*p* < 0.001) more likely to present better NB (at 3 days, patients in the protein-fortified diet were estimated to have a nitrate value of 5.22 g more than patients in the standard diet, 95% CI 3.86–6.58). The longitudinal analysis (adjusted for multiplicity) of secondary endpoints (Figure 4) confirmed the absence of significant differences between patients in the protein-fortified and standard diet.

## 4. Discussion

This randomized controlled trial aimed to compare NB in critically ill patients receiving a protein-fortified diet (as recommended by international guidelines [4]) and a standard diet, since it has been shown that the administration of standard nutrition formulae in such patients resulted in inadequate protein intake [10,19].

The study showed a positive association between a protein-fortified diet and a better NB (*p* < 0.001). Of note, according to the study protocol, patients were randomized to the standard or the protein-fortified diet within 12-h of hospital admission (Day 0). The prompt starting of the protein-fortified diet allowed for a better NB early after the ICU admission (Day 1). Therefore, consistent with the literature [20], the prompt nutritional support of critically ill patients seemed to prevent protein catabolism within the first hours of the ICU stay, without affecting the renal function.

The present results are in line with those of the trial of Ferrie et al. [5], demonstrating a better NB in the group of ICU patients undergoing amino acids supplementations (according to international guidelines). Unfortunately, clinical trials in this field are lacking. Two observational studies [8,21] have already shown that a protein or amino acids supplementation resulted in lower mortality in critically ill patients; however, they did not provide information about NB in such patients. A few studies have also tested the efficacy of a protein provision higher than that recommended. Patients with brain injuries treated with >2 g/kg/day of protein showed a better NB than those administered with 1.5 g/kg/day of protein [22]. However, even though the NB was significantly better at study exit for patients allocated to the protein-fortified diet compared to those allocated to the standard diet (Table 2), the difference in NB seemed to be lower between the two groups on day 7 and beyond (Figure 4). Several factors might have contributed to such findings, including the low number of patients observed after day 7, which were the most severe ones, since they remained in the study until mechanically ventilated. Not least, we cannot rule out that such patients required a more aggressive nutritional support (i.e., a higher protein intake).

For what concerns the secondary endpoints, the protein-fortified diet did not have any effect on renal function compared with the standard diet: creatinine level in blood was not different in the two groups. A protein supply within the range of 1.2 to 2 g/kg/day can be considered safe for critically ill patients’ renal function. Some studies suggested that providing higher protein intake can increase the risk of decreased kidney function and reduced glomerular filtration rate [23]. However, recent findings from the Cardiovascular Health Study reported that higher protein intake did not lead to an impairment of kidney function among senior men and women [24].

Skin alterations (pressure ulcers, venous ulcer, and swelling of the feet and legs -edema-) were found to be more prevalent among the standard diet group compared to the protein-fortified ones at study exit (*p* 0.013). Patients admitted to ICUs are at a higher risk of developing skin alterations than patients admitted to general care [25]. A review of literature for the period of 2000 to 2005 indicated a pressure ulcer prevalence in the ICU of 4–49% and an incidence of 3.8–40.4% [26]. The use of protein-fortified diets for patients with protein deficiency is essential to wound healing [27]; however, it is still unclear what is the optimum protein intake for patients with pressure ulcers.

### Study Limitations

The main limitation of this study is represented by the fact that patients suffering from heterogeneous clinical conditions were enrolled. Especially in patients with trauma injuries, the systemic inflammatory response lead to an increased protein catabolic rate [28]. Furthermore, individual variation in lean body mass composition was not assessed. Even though the literature described relationships among body composition, energy demand, and protein requirements [29], in this study we used only weight-based formulas. 

Not least, we have to remember that NB determination has several limitations [3]. The estimation of the nitrogen losses represented the main limitation. In the present study, we considered a constant of 4 g/day, accounting for non-urinary losses [13]. However, it is difficult to establish such a measure, since no easy measurements are available [30]. Furthermore, diet formulations and the route of administration (enteral vs. parenteral) may result in differences in absorption. These limitations can lead to an underestimation of NB and protein requirements.

Further large multicenter clinical trials might be done to address such issues (e.g., accounting for patients’ heterogeneous clinical conditions, different routes of administration of nutritional support, etc.), as has been highlighted by recent recommendations in the field [4], since the topic of optimal protein provision in ICU patients remains controversial.

## 5. Conclusions

A protein-fortified diet was found to be significantly and directly associated with NB in critically ill patients admitted to ICU. This result strengthens the rationale for the use of NB as a marker for adequate protein intake.

## Figures and Tables

**Figure 1 nutrients-11-00972-f001:**
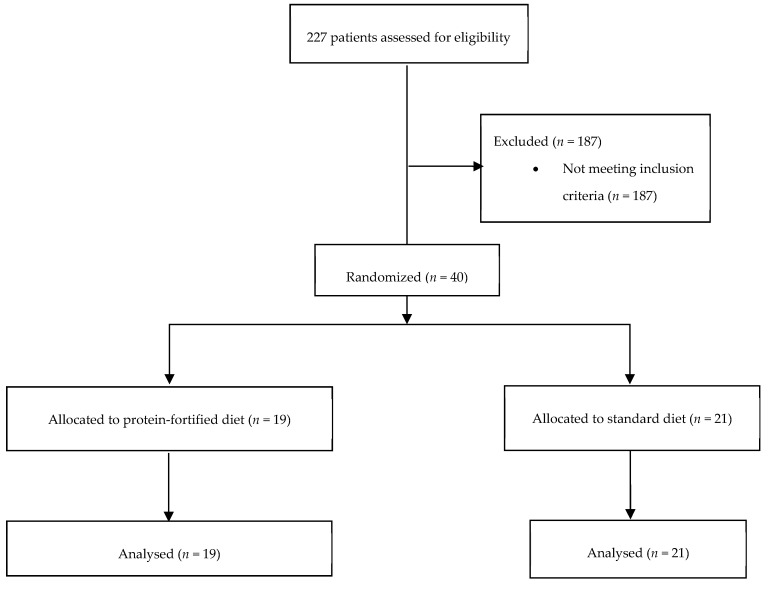
Flowchart of patient recruitment and allocation to study treatments.

**Figure 2 nutrients-11-00972-f002:**
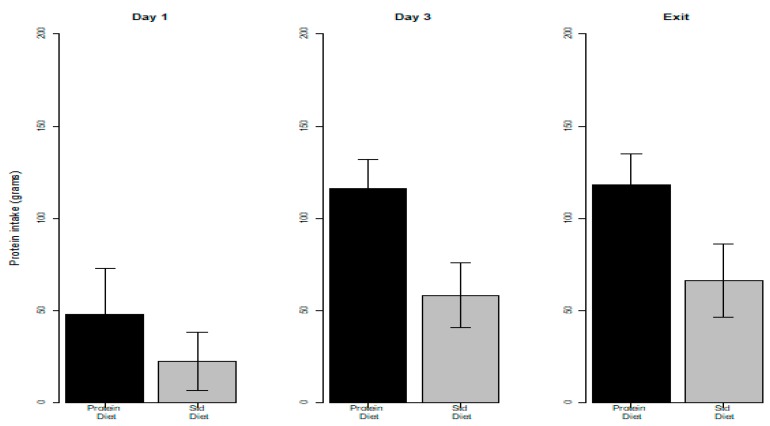
Protein intake in the intervention and control groups on day 1, 3, and study exit.

**Figure 3 nutrients-11-00972-f003:**
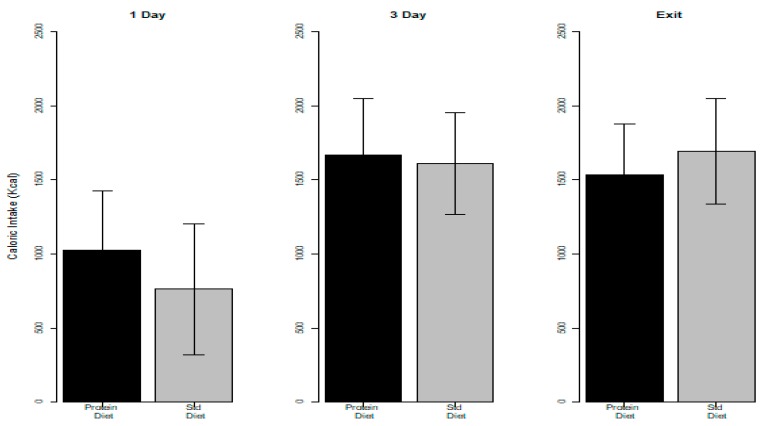
Energy (kcal) intake in the intervention and control groups on day 1, 3, and study exit.

**Figure 4 nutrients-11-00972-f004:**
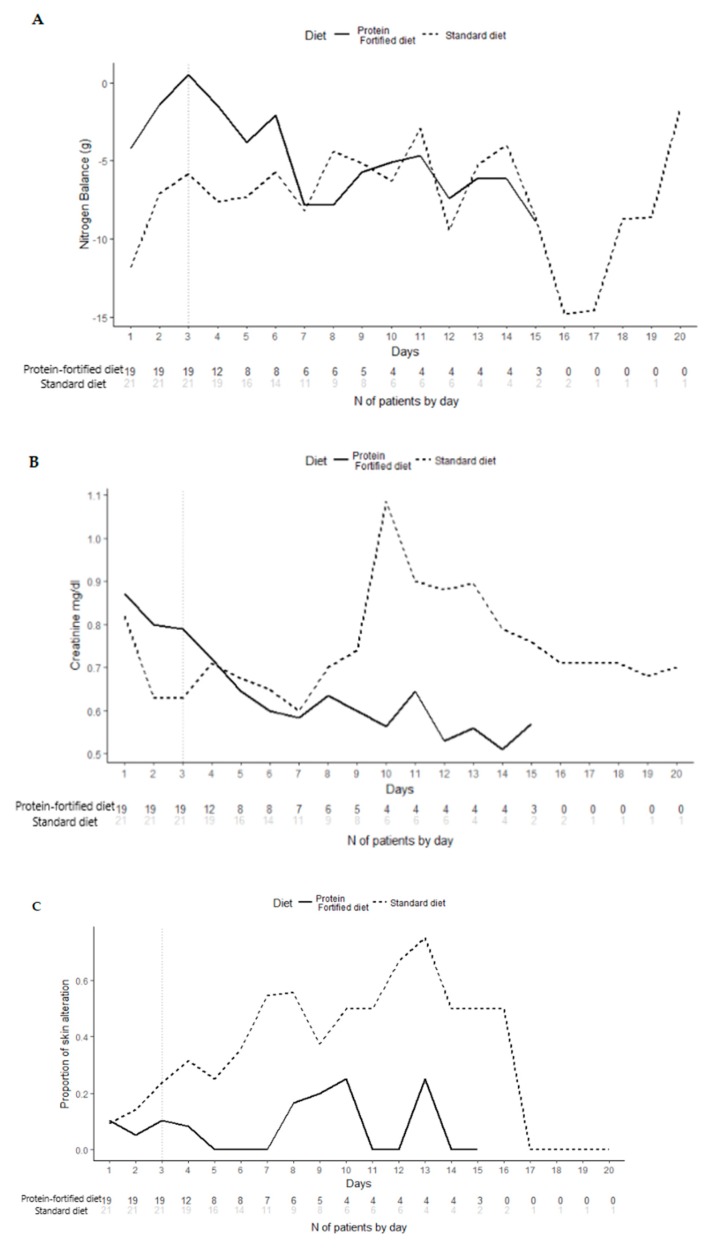
Longitudinal analysis of primary (A-nitrogen balance; *p* < 0.001. 3 Days Effect = 5.22 g 95% CI (3.86; 6.58)) and secondary (B-creatinine, *p* =0.4011. 3 Days Effect = 0.22 mg/dl 95% CI (−0.17; 0.63); C-skin alteration, *p* = 0.1676. 3 Days Odd Ratio (OR) = 0.395 95% CI (0.12; 0.76)) endpoints. Curves are the descriptive analysis of the overall follow-up. The *p*-values indicate the significance of the differences between treatment groups and have been adjusted by multiplicity for all secondary endpoints. The “3 Days Effect” refers to the model-based estimates of the differences between the two diets evaluated at 3 days (in actual units).

**Table 1 nutrients-11-00972-t001:** Characteristics of the patients at randomization (day 0).

	Protein-Fortified Diet	Standard Diet	Overall
	*n* = 19	*n* = 21	*n* = 40
Age	66.0 (57.0; 72.0)	63.0 (46.0; 70.0)	64.0 (50.5; 70.2)
Gender			
Female	42% (8)	48% (10)	45% (18)
Male	58% (11)	52% (11)	55% (22)
BMI	25.00 (23.00; 27.50)	24.00 (21.00; 26.00)	24.00 (21.75; 26.25)
Reason of admission to ICU			
Organ failure	37% (7)	43% (9)	40% (16)
Trauma	26% (5)	14% (3)	20% (8)
Cerebrovascular disease	21% (4)	33% (7)	28% (11)
Post-operative	16% (3)	10% (2)	12% (5)
Comorbidities			
Cerebrovascular insult	5% (1)	5% (1)	5% (2)
Chronic obstructive pulmonary disease	11% (2)	10% (2)	10% (4)
Coronary artery disease	21% (4)	10% (2)	15% (6)
Diabetes	26% (5)	10% (2)	18% (7)
Hypertension	26% (5)	33% (7)	30% (12)
Anxiety and/or depression disorders	5% (1)	5% (1)	5% (2)
Neoplasia	0% (0)	5% (1)	2% (1)
At least 1 comorbidity	58% (11)	48% (10)	52% (21)
Smoking habit	0% (0)	10% (2)	5% (2)
APACHE 2 score	17.00 (13.50; 22.00)	17.00 (13.00; 22.00)	17.00 (13.00; 22.00)
SAPS 2 score	32.0 (28.0; 48.5)	40.0 (31.0; 48.0)	37.0 (29.8; 48.2)
Length of mechanical ventilation	5.00 (4.00; 12.00)	8.00 (6.00; 13.00)	7.00 (5.00; 13.25)
Study exit			
Acute renal or hepatic failure	5% (1)	0% (0)	2% (1)
Death	11% (2)	33% (7)	22% (9)
Transfer to another hospital	21% (4)	14% (3)	18% (7)
Term of mechanical ventilation	63% (12)	52% (11)	57% (23)

BMI: body mass index; ICU: intensive care unit; APACHE: Acute Physiology and Chronic Health Evaluation; SAPS: Simplified Acute Physiology Score. Data are median (I quartile; III quartile) for continuous variables and percentage (number of observations) for categorical variables.

**Table 2 nutrients-11-00972-t002:** Characteristics of the patients and the distribution of main nutritional indicators during follow-up (day 1, 3, and study exit).

	Protein-Fortified Diet	Standard Diet	Overall	*p*-value
	*n* = 19	*n* = 21	*n* = 40	
Day 1				
Nitrogen balance (g)	−4.20 (−9.15; −2.20)	−11.80 (−14.10; −8.50)	−9.05 (−11.85; −4.10)	<0.001
Creatinine (mg/dl)	0.870 (0.555; 1.175)	0.820 (0.560; 1.020)	0.840 (0.558; 1.093)	0.916
ICU-acquired skin alterations	11% (2)	10% (2)	10% (4)	0.916
Day 3				
Nitrogen balance (g)	0.50 (−2.15; 1.50)	−5.80 (−10.00; −1.90)	−2.35 (−7.97; 0.55)	0.003
Creatinine (mg/dl)	0.79 (0.63; 1.03)	0.63 (0.53; 0.96)	0.75 (0.54; 1)	0.336
ICU-acquired skin alterations	11% (2)	24% (5)	18% (7)	0.336
Study exit				
Nitrogen balance (g)	0.1 (−3.4; 0.85)	−8.700 (−11.100; −5.200)	−5.15 (−9.05; 0.1)	<0.001
Creatinine (mg/dl)	0.8 (0.52; 0.99)	0.6 (0.52; 0.77)	0.69 (0.52; 0.9)	0.523
ICU-acquired skin alterations	5% (1)	38% (8)	22% (9)	0.013

Data are median (I quartile; III quartile) for continuous variables and percentage (number of observations) for categorical variables; *p*-values refer to the difference between groups and have been adjusted by multiplicity of testing.

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
