# Peer review of "Effect of Protein-Fortified Diet on Nitrogen Balance in Critically Ill Patients: Results from the OPINiB Trial"

_nutrients, 2019, doi:10.3390/nu11050972_

Round 1
Reviewer 1 Report
This was an excellent study and very timely with the controversy with protein provisions in the ICU. This study contributes greatly to the literature on protein needs. I think that the sample size is large enough to see an effect and the selected endpoints were also specific enough to have viable outcome data.
Author Response
We would like to thank the reviewer for the comments. English language has been revised.
Reviewer 2 Report
Lines 19-20: suggest specify the standard diet (25 kcal…) and protein-fortified diet (1.8 g/kg according to…suggest adding the name of the institution/guidelines that state this).
Line 56: Nitrogen loss or nitrogen balance? I suggest removing “adequately”. Instead the statement should propose that Since the influence of NB (determinants, consequences and optimal intake) in ICU patients is not clear, it is crucial to conduct further research in this field.
Line 64? Which was the maximum age for the participants allowed to participate? Age is an important factor that influence nitrogen loss and nitrogen balance in adults.
Line 66: During the intervention....Patients were in PTN or EN? Or both?, this must be stated. I guess same conditions were applied for all the patients. If there were a group of patients with PTN and other with EN differences must be considered and explained. Also, nutritional composition (carbs/dextrose, proteins, fats) of each nutrition protocol must be explained and showed. Also, if appropriate, differences in protein metabolism from EN and PTN must be stated in discussion and limitations.
Line 73: Add a cite for the “most recent guidelines”
Lines 74-77: why did you use different formulas for calculating energy requirements? The REE can be applied to the standard diet group? Add a reference for the REE. The energy goal for the standard diet was settled at 20-25 Kcal/kg/day, but the gr/kg of protein or the amount of protein was how much?
Line 83: how did you measure height, weight...please specify the equipment used, and also weight measurement conditions (if they were in fasted state, bedridden, possible edema?)…
Line 85: please specify the methods/equipment/ laboratory kits you used for measuring blood test variables…
Line 102: I am not sure if you have mentioned but I did not see which in statistical analysis which covariables did you use for adjustments? Did you consider medications, age, sex or something else?...Some medications like diuretics or AINES could affect protein excretion in urine.
Line 125: I would suggest to better explain this table. Which aspects would you highlight?
Line 161: Suggest, this randomized controlled trial aimed TO COMPARE NB in critically ill patients….
Line 165: Suggest, the study showed a positive association between a protein-fortified diet and a better NB (p-value <0.001).
Lines 175-181: suggest moving this paragraph to limitations section.
Line 197: please consider if necessary, adding the above mentioned. Also consider adressing the influence of diabetes in renal function and differences in the nutrition method (EN or PTN) influencing protein absorption.
TABLE 1:
I see unnecessary the column “N” considering that subjects (n) in each group (standard and protein fortified diet are specified and also total (overall n=40)).
Were all the subjects females? According to table 1, only females were listed and a n=40 is listed. It´s confusing to understand. However, I see that that finally 18, were females. I suggest removing the n=40 column. Also, I suggest to specify the number (%) males for each column.
BMI have not been mentioned before, please add in the method section, defining the acronym and citing the original reference (Obesity : Preventing and Managing the Global Epidemic….Report of WHO consultation.2000).
About subjects with diabetes, which type of diabetes? Could it affect renal function in this group of patients? Please explain this.
APACHE 2 and SAPS 2: for a better reader interpretation please define these acronyms in footnotes.
How did you measure smoking habit? Please add in the method section.
FIGURES
For both figures the unit of measurement must be specified. Protein (grams) ? Calories (kcal)?
TABLE 2
Title: Is it day 2 and 3 or day 1 and 3?
4, 0 in the N column... I guess is...40, however same as in table 1, I suggest to remove this column.
Did you adjust by something? Which covariables did you use? Please specify this in footnotes.
TABLE 3
Is it a table or a figure?
Author Response
Comments and Suggestions for Authors
1) Lines 19-20: suggest specify the standard diet (25 kcal…) and protein-fortified diet (1.8 g/kg according to…suggest adding the name of the institution/guidelines that state this).
Done
2) Line 56: Nitrogen loss or nitrogen balance? I suggest removing “adequately”. Instead the statement should propose that Since the influence of NB (determinants, consequences and optimal intake) in ICU patients is not clear, it is crucial to conduct further research in this field.
Done.
3) Line 64? Which was the maximum age for the participants allowed to participate? Age is an important factor that influence nitrogen loss and nitrogen balance in adults.
The study enrolled subjects aged >18 years. No differences were found in age distribution of subjects randomized to the protein-fortified and standard diet.
4) Line 66: During the intervention....Patients were in PTN or EN? Or both?, this must be stated. I guess same conditions were applied for all the patients. If there were a group of patients with PTN and other with EN differences must be considered and explained. Also, nutritional composition (carbs/dextrose, proteins, fats) of each nutrition protocol must be explained and showed. Also, if appropriate, differences in protein metabolism from EN and PTN must be stated in discussion and limitations.
Each patient enrolled in the study could undergo to both enteral and/or parental nutrition according to the clinical judgement and guidelines in the field. The nutritional support (enteral and/or parenteral) was set up in order to comply with the energy requirements of each subject. The nutritional composition of enteral nutrition and parenteral nutrition of patients in the standard diet was as follows: protein (16%), carbs/dextrose (35%), fats (49%) and protein (16%), carbs/dextrose (49%), fats (35%), respectively. The nutritional composition of enteral nutrition and parenteral nutrition of patients in the protein-fortified diet was as follows: protein (21%), carbs/dextrose (31%), fats (46%), fiber (2%) and protein (22%), carbs/dextrose (40%), fats (38%), respectively. The proportion of protein administered to the subjects allocated to the protein-fortified diet was specific for each patient according to her/his energy requirements, the ones reported is an average value. The clarification requested has been added to the Methods and Limitations section.
5) Line 73: Add a cite for the “most recent guidelines”
Done.
6) Lines 74-77: why did you use different formulas for calculating energy requirements? The REE can be applied to the standard diet group? Add a reference for the REE. The energy goal for the standard diet was settled at 20-25 Kcal/kg/day, but the gr/kg of protein or the amount of protein was how much?
According to the current guidelines, the standard diet consisted in an energy goal settled at 20-25 Kcal/kg/day, while the protein-fortified diet consisted of an energy goal based on REE. The amount of protein was 1.8/g/kg/day.
7) Line 83: how did you measure height, weight...please specify the equipment used, and also weight measurement conditions (if they were in fasted state, bedridden, possible edema?)…
Done
8) Line 85: please specify the methods/equipment/ laboratory kits you used for measuring blood test variables…
The information requested has been added to the manuscript.
9) Line 102: I am not sure if you have mentioned but I did not see which in statistical analysis which covariables did you use for adjustments? Did you consider medications, age, sex or something else?...Some medications like diuretics or AINES could affect protein excretion in urine.
A randomized controlled trial was done. No significant differences were found in any one of the characteristics of the study subjects randomized to the standard and protein-fortified diet. The randomization resulted in a balanced allocation of the subjects in the two groups. For this reason, no adjustments were made.
10) Line 125: I would suggest to better explain this table. Which aspects would you highlight?
The models were estimated by including an interaction term between the treatment (standard vs. protein-fortified diet) and time to account for potential different time-dependent patterns of the endpoint of interest in the control and in the experimental group. Clarifications have been added to the Methods section.
11) Line 161: Suggest, this randomized controlled trial aimed TO COMPARE NB in critically ill patients….
Done
12) Line 165: Suggest, the study showed a positive association between a protein-fortified diet and a better NB (p-value <0.001).
Done
13) Lines 175-181: suggest moving this paragraph to limitations section.
Done
14) Line 197: please consider if necessary, adding the above mentioned. Also consider adressing the influence of diabetes in renal function and differences in the nutrition method (EN or PTN) influencing protein absorption.
Done. Clarifications have been added to the limitation section.
TABLE 1:
15) I see unnecessary the column “N” considering that subjects (n) in each group (standard and protein fortified diet are specified and also total (overall n=40)).
Done
16) Were all the subjects females? According to table 1, only females were listed and a n=40 is listed. It´s confusing to understand. However, I see that that finally 18, were females. I suggest removing the n=40 column. Also, I suggest to specify the number (%) males for each column.
The number of males has been specified
17) BMI have not been mentioned before, please add in the method section, defining the acronym and citing the original reference (Obesity : Preventing and Managing the Global Epidemic….Report of WHO consultation.2000).
Done
18) About subjects with diabetes, which type of diabetes? Could it affect renal function in this group of patients? Please explain this.
Insulin-dependent diabetes. The renal function was strictly monitored daily. The protein-fortified diet appeared to be effective without affecting renal function and no differences were found in renal function in the protein-fortified and standard diet.
19) APACHE 2 and SAPS 2: for a better reader interpretation please define these acronyms in footnotes.
Done
20) How did you measure smoking habit? Please add in the method section.
Done
FIGURES
21) For both figures the unit of measurement must be specified. Protein (grams) ? Calories (kcal)?
Done. The unit of measurement was added to the Figures.
TABLE 2
22) Title: Is it day 2 and 3 or day 1 and 3?
There was a mistake in the Table caption. It is Day 1, 3 and Study exit. The mistake has been fixed.
23) 4, 0 in the N column... I guess is...40, however same as in table 1, I suggest to remove this column.
Done
24) Did you adjust by something? Which covariables did you use? Please specify this in footnotes.
No adjustments were made.
TABLE 3
25) Is it a table or a figure?
It is a Table